# Activation-inducible CAR expression enables precise control over engineered CAR T cell function

Simon P. Fraessle[1,3], Claudia Tschulik[1,3], Manuel Effenberger [1,3✉], Vlad Cletiu[1], Maria Gerget[1], Kilian Schober [2], Dirk H. Busch [2], Lothar Germeroth[1], Christian Stemberger[1] & Mateusz P. Poltorak[1]

CAR T cell therapy is a rapidly growing area of oncological treatments having a potential of becoming standard care for multiple indications. Coincidently, CRISPR/Cas gene-editing technology is entering next-generation CAR T cell product manufacturing with the promise of more precise and more controllable cell modification methodology. The intersection of these medical and molecular advancements creates an opportunity for completely new ways of designing engineered cells to help overcome current limitations of cell therapy. In this manuscript we present proof-of-concept data for an engineered feedback loop. We manufactured activation-inducible CAR T cells with the help of CRISPR-mediated targeted integration. This new type of engineered T cells expresses the CAR gene dependent on their activation status. This artifice opens new possibilities to regulate CAR T cell function both in vitro and in vivo. We believe that such a physiological control system can be a powerful addition to the currently available toolbox of next-generation CAR constructs.

[1] Juno Therapeutics GmbH, a Bristol-Myers Squibb Company, Grillparzerstr. 10, 81675 Munich, Germany. [2] Institute for Medical Microbiology Immunology and Hygiene, Technical University of Munich, Munich, Germany. [3]These authors contributed equally: Simon P. Fraessle, Claudia Tschulik and Manuel Effenberger. ✉email: manuel.effenberger@bms.com

Genetically engineered T cells expressing modified T Cell Receptor (TCR) or Chimeric Antigen Receptor (CAR) are powerful tools to treat tumor malignancies[1]. Until today, several CAR T cells therapies against hematological disorders have been approved with a plethora of others being evaluated in clinical trials[2]. All the current commercial products use viral vector-based gene delivery[3]. Although this is a well-established and relatively safe strategy, it possesses some significant limitations. Mechanistically, contemporary viral vectors rely on uncontrolled semi-random integration. Although safety of viral vectors is proven by the increasing number of patients treated with CAR T cell products and safety of viral gene delivery has been accepted by health agencies, the inherent risk of insertional mutagenesis remains an unsolved issue[4–6]. Therefore, next-generation T cell manufacturing aims to use better controlled gene editing methods such as CRISPR/Cas. CRISPR/Cas enables precise targeted gene integration into pre-defined location within the genome[7]. This in turn not only makes the editing procedure more precise but also opens an opportunity to take advantage of the gene integration site.

To date all clinically used CAR gene cassettes are driven by strong, constitutively active artificial promoters, overwriting the dynamic transcriptional regulation that occurs in a physiological setting with direct impact on activation kinetics, phenotype, and survival[8]. Continuous CAR expression can lead to exhaustion and suboptimal CAR functionality in vivo[9]. The CRISPR/Cas technology allows gene transfer downstream of any endogenous promoter, thus reducing the need of engineered or external promoters and focus on more physiologic cell modifications[8,10]. Native promoter regions enable highly regulated gene transcription. The complex sequence patterns funnel regulatory cues from distal enhancers and their associated modulating proteins lead to distinct transcription[11]. Preserving the sophisticated physiological gene regulation may help to engineer differentiated cell products. It has been demonstrated that better targeting of CARs and engineered TCRs brings clear benefits to the potency and function of edited T cells by inserting artificial genes under control of endogenous TCR promoters thus providing a more physiological (trans-)gene expression[12].

Moreover, a well-designed editing strategy can enhance precision of CAR-mediated killing. CAR T cells are very potent in the lysis of cells expressing target antigen (e.g., CD7, CD19 or BCMA), but they do not distinguish between healthy and abnormal cells[13–15]. Thus, after performing their initial task of tumor clearance, CAR T cells continue to attack all cells expressing their cognate antigen[16]. Several approaches to circumvent these on-target off-tumor side effects of persisting CAR T cells have been proposed. For example, CAR T cells can be eradicated after complete tumor clearance by systemic administration of antibodies directed against a synthetic "kill switch" co-expressed on CAR T cells, leading to complete and persistent B cell recovery[17]. Another option is mRNA-based transient expression of CAR, which re-directs T cells against target cells only as long as mRNA is systemically supplied[18,19]. However, such approaches have challenges of addressing distribution kinetics, precision, penetrance, and persistence of applied molecules.

Another method of reducing damage to healthy tissue is logically gated CAR T cells[20]. For example, 'AND' gated CAR T cells can differentiate tumor cells from healthy tissue more precisely since they only execute effector function by dual antigen recognition on target cells[21]. However, healthy tissue destruction is only securely prevented if both antigens are spatially separated enough to prevent dual recognition. In addition, 'NOT' and 'OR' gated CARs were also described in preclinical models[22]. These logic gates seem to be a very appealing and sophisticated approach to regulate edited T cell function but require complex cell engineering and the genetic payloads are bulky[23].

A more elegant method was recently described where CRISPR-mediated knock-in of IL-15 was directed into the inducible locus of IL-13[24]. The authors observed a significant although moderate increase of IL-15 production upon cell stimulation validating their concept but highlighting the importance of adequate locus selection.

In this article, we propose an expansion of the targeted approach to solve the issue of persistent on-target off-tumor tissue destruction after tumor clearance using control over TCR/CAR gene expression. Our strategy aims to reduce the necessary genetic cargo and relies instead on physiological regulation of the engineered protein. To this end, we propose the use of constructs which are embedded at loci of molecules that are temporarily and strongly expressed via physiological signal cascades. In the case of T cells, apparent target loci are downstream of promoters that upregulate their cognate proteins upon T cell activation (e.g.: Nur77, FoxP3, CD69, PD-1, HLA-DR)[25,26]. With this approach we created a temporal regulated system where persistence of T cell stimulation (exogenous or endogenous) is prerequisite to exert CAR/TCR function. We call it an engineered feedback loop: when cells are active, target construct is expressed. In this study, we demonstrated feasibility through in vitro and in vivo proof-of-concept experiments. We are convinced that our engineered feedback loop can be utilized in advanced cell therapy applications.

## Results

**CAR expression can be linked to T cell activation.** We conceptualized that theoretically any construct can be regulated in a similar fashion to endogenous up- or downregulation if it is targeted into an appropriate locus and inserted via homologous recombination. As a proof-of-concept example, we selected anti-CD19 CAR T cells as an available and relevant setting for therapeutic T cell engineering (Fig. 1a). In this case, the CAR construct is integrated into the area downstream of a T cell activation-dependent promoter, simultaneously disrupting the coding sequence of the endogenously coded protein. Of note, for intended evaluations, TCR/CD3 complex was left unedited, enabling classical anti-CD3/CD28 T cell (re)stimulation that is independent of anti-CD19 antigen-specific activation. In our example, T cells are first polyclonally stimulated and then modified with the CRISPR/Cas9 system, whereby the CAR gene is integrated into the specific gene locus substituting the downstream sequence with the CAR (Fig. 1a). Our editing strategy follows conventional CRIPSR protocols where editing occurs 24–48 h post-activation and it is in line with state-of-the-art CAR T cell manufacturing processes (e.g., ultra-short processes) that require initial polyclonal stimulation. As long as this initial cell activation is maintained (especially in an antigen-specific setting) the CAR continues to be expressed creating a positive feedback loop (activation via CAR target recognition facilitates further CAR protein production) (Fig. 1a). This engineered feedback loop should allow control of CAR expression by the cell activation state.

To test our hypothesis, we evaluated several well-known markers that are upregulated upon T cell stimulation (Fig. 1b). As selection criteria, we assessed expression kinetics, the restimulation potential as well as biological significance. We initially focused on two classical T cell activation markers—CD25 and CD69—but also tested upregulation of PD-1, TIGIT and TIM3 markers. These latter receptors are not only upregulated upon T cell stimulation, but their prolonged presence is correlated with T cell exhaustion[26]. For the initial evaluation, CD25 turned out to

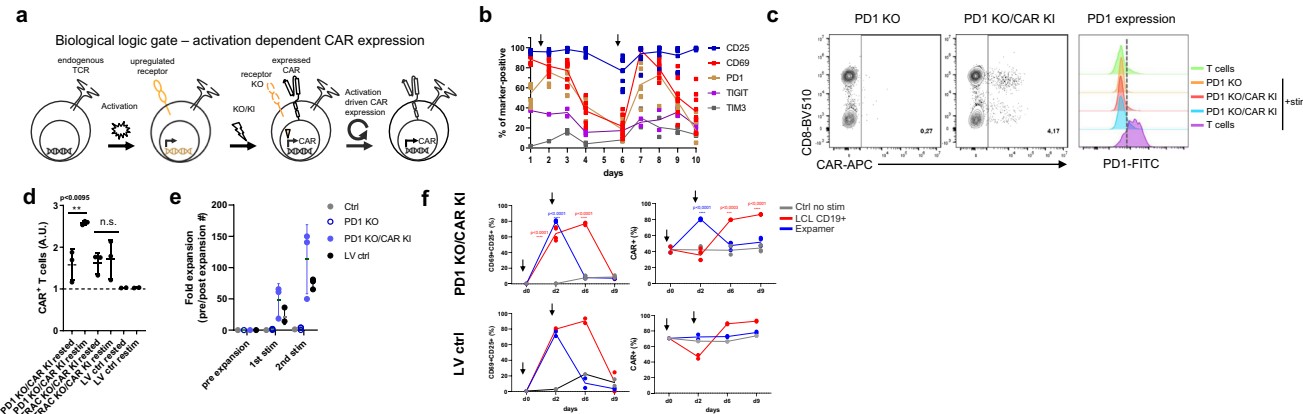

**Fig. 1 CAR expression can be linked to T cell activation. a** Schematic representation of CAR expression under control of activation-dependent promoter. **b** Physiological expression kinetics of PD1, CD25, CD69, TIGIT, and TIM3 activation markers. Graph presents marker expression over time from at least 3 donors. Cells were restimulated on d6 (arrow) after initial stimulation after d1 . Connecting lines represent mean. **c** CAR can be successfully knocked-in into PD1 locus. Set of representative dot plots and histogram are shown. Plots pre-gated on living, single CD45+ cells. **d** CAR expression under PD1 locus is activation-dependent upon polyclonal anti-CD3/CD28 stimulation. Graphs display CAR expression as arbitrary units (A.U.) in rested or restimulated samples from 3 independent experiments. CAR constructs targeted into TRAC locus as well as expressed via lentiviral (LV) random integration were used as a control. Error bars represent mean ± SD. The significance was analyzed using unpaired Student's *t*-test. **e** CAR T cells can be expanded via antigen-specific stimulation. Graph depicts fold expansion upon two rounds of antigen-specific stimulation. Errors bars represent mean ± SD. **f** Antigen-specific expansion leads to increased CAR T cell content in all samples, whereas polyclonal restimulation enhances CAR T cell content in activation-induced CAR expression setting only. Graphs display indicated marker expression as frequencies in rested or restimulated samples from 3 independent experiments. CAR constructs targeted into PD1 locus as well as expressed via lentiviral (LV) random integration are depicted. Arrows indicate start (d0) and end (d2) of stimulation period. Connecting lines represent mean. The significance was analyzed using paired Student's *t*-test.

be unsuitable for our approach since its expression was elevated even throughout resting phase (Fig. 1b). Likely, TIGIT and TIM3 were excluded because their expression was below 50% after stimulation (Fig. 1b). Ultimately, we selected PD-1, as its function is best characterized and PD-1 inhibition is a broadly utilized technique in check-point inhibitor treatments[27]. Theoretically, disruption of PD-1 can provide benefits for CAR T cells as previously described[28,29]. Hence, for the first experiments we focused on CAR targeting into the PD-1 locus to establish a 2-in-1 model, in which we combine engineered feedback loop CAR expression with the potential benefit of PD-1 knock-out (KO). In addition, to demonstrate broader application of our concept we also tested targeting of CD69 locus (Supplementary Fig. 1a-b). In our model we used human primary T cells stimulated with soluble anti-CD3/CD28 reagent (Expamers) and used a non-viral CRISPR/Cas9 delivery system and dsDNA HDR template as previously described[8]. Edited T cells readily expressed the CAR construct but had almost no residual expression of PD-1, indicating successful targeted integration as well as very efficient PD-1 KO (Fig. 1c, Supplementary Fig. 2). These cells were then either left unstimulated to reach the resting phase or restimulated with an additional dose of Expamers. 24 h post-restimulation, CAR expression was evaluated (Fig. 1d). Conversely to the controls (CAR gene knock-in (KI) into TRAC locus and CAR randomly integrated by LV transduction), expression of CAR construct located downstream of the PD-1 promoter significantly increased after restimulation (Fig. 1d). This was the first experimental validation that engineered feedback loop CAR expression can be achieved through our approach. Interestingly, CAR expression in CRISPR/Cas edited cells seems to increase in vitro spontaneously over time (and irrespectively of activation state) in contrast to LV-mediated randomly integrated receptors (Fig. 1d) again pointing to the incomplete PD-1 downregulation in resting phase in this in vitro setting (Fig. 1b) and might hint to an improved fitness of CAR engineered T cells.

Next, we decided to test the robustness of our inducible CAR concept over an extended period of two rounds of antigen-

specific stimulation (Fig. 1e). We anticipated that antigen-specific stimulation will increase relative frequency of anti-CD19 CAR T cell population within the samples irrespective of type of editing used. For this purpose, we expanded CAR T cells using B-cell-derived Lymphoblastoid Cell Line (LCL), which expresses CD19 target antigen. After 1 week of co-culture, we indeed observed significant cell expansion in all samples of interest indicating productive antigen-specific cell responses. Interestingly, fold-increase in cell expansion was higher in T cells with CAR controlled by the PD-1 promoter compared to the LV control (Fig. 1e). To further examine maintenance of the initially reported behavior of engineered feedback loop CAR T cells and to improve cellular resolution of the assays, we assessed the response of LCL-expanded cells under polyclonal and antigen-specific restimulation (Fig. 1f). In particular, we evaluated whether dynamic upregulation of CD25 and CD69 activation markers matches anticipated kinetic of engineered feedback loop CAR expression. As expected, after polyclonal restimulation both CD25/CD69 and CAR content increased significantly at day 2 post restimulation and returned to baseline levels at day 6 (Fig. 1f, blue line). Noteworthy, only in the group of activation-inducible CAR T cells increase in activation markers correlated with transient rise of CAR expression. This contrasted with the antigen-specific restimulation, where enhanced activation corresponded to higher CAR content indicating expansion of CAR+ cells upon target engagement irrespectively of CAR integration strategy whereby activation markers are upregulated until day 6 reaching baseline later on day 9 after complete target cell elimination (Fig. 1f, red line). Of note, initial decrease in CAR+ frequency upon antigen-specific restimulation can be accounted to early interactions between CAR+ and target cells leading to steric hindrance and CAR epitope masking. These data confirmed that proposed engineered feedback loop does not diminish over time and becomes a permanent feature of edited cells. Of note, during all our studies we observed heightened baseline level of CAR (~40%) (Fig. 1f) over steady state PD-1 presence (( < 20%) Fig. 1b). This may be the outcome of editing

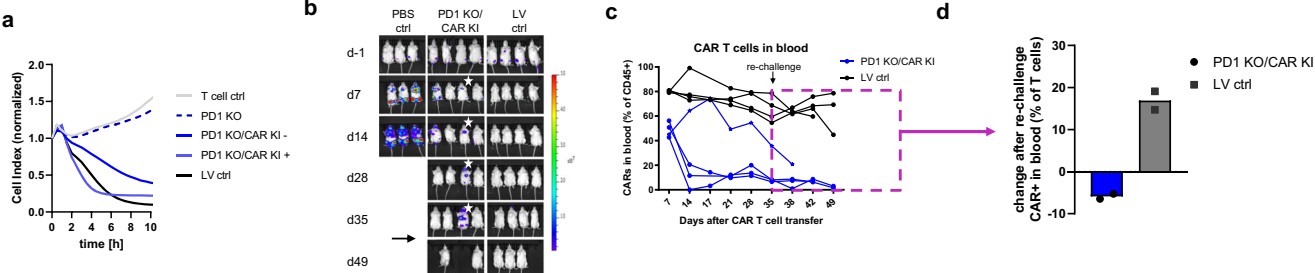

**Fig. 2 Inducible CAR T cells are functional in vitro and in vivo. a** Activated engineered feedback loop CAR T cells show faster in vitro killing compared to rested counterpart. Graph shows impedance measurement as a read out of antigen specific killing of target cells. Untransduced T cells (neg control), PD1 KO only T cells (PD1 KO), CAR T cells LV-transduced (LV control) as well as rested (PD1 KO/CAR KI -) and polyclonally restimulated (PD1 KO/CAR KI +) engineered feedback loop CAR T cells were used. Experiment shows single data points derived from median of quadruplicates. **b** Engineered feedback loop CAR T cells are as functional as state-of-the-art CAR T cell product in an in vivo mouse model. Tumor was detected using IVIS bioluminescence measurements. Luminescence images from one representative experiment are shown. Arrow indicates re-challenge of mice. **c** Engineered feedback loop CAR T cells can help restore B cell compartment after complete tumor eradication. CAR T cell content in blood of each individual mouse was calculated using cell frequencies measured via flow-cytometry. Cells were pre-gated on living lymphocytes. Box highlights cell frequencies after B cell re-challenge (d35). Mouse with prolonged tumor burden highlighted by star in **b** and **c**. Each line represents corresponding mouse. **d** B cell re-challenge on day 35 leads to rapid increase in LV-transduced CAR T cell content while engineered feedback loop CAR T cell content remains unchanged. Delta calculated based of CAR T cell frequencies between day 28 (tumor clearance) and day 45 (post re-challenge).

strategy and an impact of PD-1 KO on general T cell biology or an artifact of the in vitro culture.

**Inducible CAR T cells are functional in vitro and in vivo.** After verifying activation-inducible CAR expression in practical experiments, we decided to test the functionality of generated CAR T cells and potential benefits of additional levels of control. To this end, we performed both in vitro and in vivo killing assays (Fig. 2a, b). For the in vitro killing assay, we prepared engineered feedback loop CAR T cells which were either activated (PD1 KO CAR KI +) or left unstimulated (PD1 KO CAR KI -) to gauge their cytolytic potency (Fig. 2a). As anticipated, stimulation boosted killing function of CAR T cells measured as faster clearance of CD19 + HEK target cells when compared to unstimulated control. Importantly, this result can be interpreted as a lowered activation state that will translate into reduced potency. Both observations are in line with the concept that antigen-mediated function can be regulated by cell activation state. Afterwards, we assessed cell killing efficacy in a xenograft Raji mouse model (Fig. 2b). As presented, produced CAR T cells effectively cleared tumor cells within few weeks in a comparable manner regardless of CAR integration strategy. Noteworthy, also T cells expressing CAR under CD69 promoter demonstrated tumor control, albeit less potent than under PD-1 (Supplementary Fig. 1c–e). These data showcase the capability not only of PD1 KO CAR KI but also the general approach of CRISPR-mediated gene engineering.

Based on the in vitro observation that the activation state may translate into CAR T cell potency on a population level, we expected to recapitulate similar effects in vivo. We hypothesized that re-challenge of mice few weeks after tumor clearance should lead to at least delayed response of activation-induced CAR T cells when compared to constantly CAR expressing counterparts (Fig. 2c, d). To this end, we injected primary B cells derived from the same donor as CAR T cells to mimic a scenario where healthy B cells can be reinstated in patient's bloodstream after CAR-mediated tumor remission. Although we were not able to detect B cells directly at any time point due to limitation of our mouse model, we observed significant difference in tested CAR T cell populations over time. First, we observed fairly constant high frequencies of CAR+ cells that were generated by random integration even 5 weeks after initial injection and 3–4 weeks after tumor removal (Fig. 2c). In contrast, engineered feedback loop

CAR T cells content dramatically decreased after 2 weeks, which corresponded with tumor absence. Only in one outlier animal, both CAR T cell maintenance and tumor persistence were proportionally prolonged (Fig. 2b, c). Interestingly, upon B cell re-challenge (d35) we observed a significant increase in conventional CAR T cells conversely to activation-inducible CAR T cells suggesting a more rapid antigen-specific response of cells with constitutive CAR expression (Fig. 2d). The observed reaction is most likely due to the much higher content of former cells in the steady state. This finding is in line with our hypothesis that continuous unaltered presence of CAR T cells in the circulation will inhibit natural B cell restoration. Unfortunately, our studies could not be prolonged beyond 7 weeks as the animals started to exhibit GvHD symptoms and had to be sacrificed due to the xenograft nature of used mouse model.

In conclusion, we demonstrated through in vitro and in vivo proof-of-concept experiments that CAR expression under activation-inducible endogenous promoter can result in regulated functional CAR T cells.

**Discussion**
In this manuscript we present an approach to control exogenous construct expression, and thus gained function of edited cells, via endogenous gene regulation in a form of an engineered feedback loop. In this setting, an artificial gene coding for construct-of-interest is introduced into the genome at the position that is managed by a promoter, whose activity depends on cellular activation state. In the proof-of-concept experiments we focused on a clinically relevant scenario and utilized a CAR construct for T cell modification. We used several significantly upregulated classical activation-dependent T cell markers to test our hypothesis such as, CD25, CD69, TIM3, TIGIT, and PD-1. In this case, the CAR construct can be co-expressed (e.g.: CD69) or disrupt the targeted gene (e.g.: PD-1), thereby theoretically improving anti-tumor potency depending on what is more desirable. Another engineering approach for physiologic expression of an artificial gene might be integration downstream of receptors that are downregulated during exhaustion, such as DNAM-1. This strategy could prevent terminal T cell exhaustion, thus increasing functional T cell persistence[30,31]. In the presented data, we targeted and disrupted PD-1 as well as CD69 genes. However, we did not evaluate to sufficient extend any previously reported benefits of PD-1 disruption.

In this activation-inducible system we were able to demonstrate that CAR content can be regulated on the population level by the cellular activation state. Moreover, this control follows the physiologic expression pattern of targeted proteins and upregulation dynamics upon indirect (non-CAR-mediated) stimulation. We demonstrated that this pattern can be maintained for at least three rounds of stimulation. The only detected divergence from physiological state was much higher frequency of CAR-positive T cells in unstimulated state. This higher-than-expected basal expression might be explained by tonic signaling through the CAR, albeit a non-clustering CAR was chosen. It has been described previously that ligand independent CAR oligomerization could drive T cells into exhaustion[32]. Alternatively, elevated expression levels in steady state can be interpreted as artifacts of in vitro culture and culture-induced unspecific activation. Interestingly, in in vivo mouse models, CAR-positive T cell frequencies were resembling more closely the natural expression kinetics of targeted activation markers since CAR frequency declined in parallel to the tumor eradication and the anticipated physiological downregulation of PD-1.

The control over CAR content translated into different qualities of the function of T cell populations both in vitro and in vivo. In the in vitro killing assay, re-stimulated cells cleared antigen-expressing cells substantially faster than rested ones. Although we cannot formally exclude the possibility, we do not expect this to be an outcome of restimulation-induced phenotypic shift due to short time frame between the conditions. Interestingly, in the Raji mouse model, we observed a pronounced correlation between T cell activity during tumor clearance and proportion of CAR-positive T cells detected in the blood. This correlation was even visible on the single-animal level where a mouse with longer persistence of malignant cells had high CAR T cell levels for prolonged period. However, in all mice that received the PD1 KO CAR KI cells, CAR frequency decreased substantially after tumor elimination in contrast to control animals receiving CAR T cells generated by lentiviral transduction. As a consequence, upon B cell injection, lentiviral control mice responded much faster as measured by a significant spike in CAR content. Nevertheless, we need to state that our measurements focused on CAR T cell frequency in the blood, and we cannot exclude migration of these cells into other body niches.

These results imply a potential application of the engineered feedback loop as a temporal safety mechanism for CAR T cell therapies. If the construct knock-in is combined with simultaneous disruption of endogenous TCR, T cell activation (induced with initial stimulation during manufacturing) will be maintained exclusively by CAR function and will cease upon depletion of all target cells in vivo. In other words, as long as CAR T cells find their target cells (e.g.: CD19 expressing tumor cells) they will remain active and perform their functions. After removal of all target cells, T cell activation and biologically associated CAR expression will decline. This will eventually render the CAR T cells unresponsive allowing healthy cell populations presenting same target antigen to recover (e.g.: healthy B or T cell compartment). Additionally, lack of endogenous TCR will prevent cell reactivation. Unresponsive CAR-TCR double-negative T cells should not affect patient's recovery and may die off due to their inability to receive tonic signaling. However, to fully validate this hypothesis further studies need to be performed, such as chronic restimulation assays.

This contrasts with classical LV-mediated random integration that drives continuous expression of CAR on T cell surface. Such CAR T cells are most likely to respond faster to a potential relapse and better control tumor recurrence. Therefore, we do not dispute potential of conventionally edited CAR T cells, but rather offer a new editing mechanism, which can be tailored depending on the target indication. Additionally, we tested our concept only on readily available CAR construct (anti-CD19). It would be still interesting to test our CAR design under different circumstances, such as lower antigen densities targets (e.g., HER2 or GPC2) and if this low antigen density has to be counterbalanced by the use of stronger endogenous promotor and/or high affinity CAR designs.

We believe that the proposed editing logic can also be exploited in other clinically relevant scenarios in which on-target off-tumor killing makes temporal limited cell therapies desirable. The presented engineered feedback loop is a powerful tool combining state-of-the-art gene delivery methods to further refine T cell therapies. We hope our work will inspire further developments in the field of CAR T cells.

## Methods

**Primary cells**. PBMCs were either isolated from fresh buffy coats via density gradient centrifugation using Biocoll (Biochrom) or selected from a Leukapheresis material using in house T cell selection technology ATC. Buffy coats were obtained from autologous adult female or male blood donors at the Institute for Anesthesiology, German heart center Munich (State of Bavaria and Technical University Munich). Written informed consent was obtained from each donor and usage of blood samples was approved according to national law by the local Institutional Review board and the declaration of Helsinki and Istanbul (Ethics committee of the faculty of Medicine, Technical University Munich: 360/13 and 55/14). Leukapheresis samples received from healthy donors were collected at the CCC Cellex Collection Center Dresden under the ethical quote (Ethical committee of the Technical University Dresden: EK309072016).

**T cell culture and activation**. Generally, T cells were enriched using in-house T cell selection technology[33]. Isolated T cells were cultured in serum free Medium (Thermo Fisher) in the presence or absence of cytokines. Enriched T cells were stimulated at specified time point with Expamers as previously described[30].

**Generation of CRISPR/Cas reagents**. crRNA sequences for gRNAs were 5'-CGTCTGGGCGGTGCTACAACGTTT-TAGAGCTATGCT-3' for PDCD1 (targeting PDCD1) and 5'-AGAGTCTCTCAGCTGGTACAGTTTTAGAGCTA TGCT-3' for TRAC. RNPs were prepared according to manufacturers' recommendation. Briefly, 80 μM tracrRNA (IDT DNA) and 80 μM crRNA (IDT DNA) were incubated at 95 °C for 5 min, then cooled to RT. 24 μM high fidelity Cas9 (IDT DNA) was added slowly to gRNA solution to yield RNPs with 12 μM Cas9 and 20 μM gRNA, as well as 20 μM electroporation enhancer (IDT DNA). RNPs were incubated for 15 min at RT and directly used for electroporation.

Double-stranded DNA PCR products were used for electroporation and HDR (Homology-directed repair). Plasmid DNA was amplified by PCR according to the following manufacturers' protocol. Thermal cycling was performed as following: 95 °C and 30 s for initial denaturation followed by 34 cycles of 95 °C for 30 s, 62 °C for 30 s, 72 °C for 3 min and 72 °C for 4 min as final extension step. Final PCR products were purified by Ampure XP beads according to manufacturer's protocol (Beckman Coulter). All HDR templates were titrated. Generally, electroporation of 1 μg DNA per $1 \times 10^6$ cells yielded best knock-in efficiencies.

**Gene editing**. Purified T cells were activated for 48 h with Expamers and 300 IU ml$^{-1}$ recombinant human IL-2, 5 ng ml$^{-1}$ recombinant human IL-7 (Peprotech) and 5 ng ml$^{-1}$ IL-15. Expamer stimulus was disrupted by incubation with 1 mM D-biotin (Sigma). $1 \times 10^6$ cells were electroporated (pulse code EH100) with Cas9 ribonucleoprotein and DNA templates in 20 ul Nucleofector Solution P3 (Lonza) with a 4D Nucleofector X unit (Lonza). After electroporation, cells were cultured in serum free media with 180 IU ml$^{-1}$ IL-2 until first analysis on day 5 after editing. KI efficiency was measured using and anti-CAR antibody specific against the introduced scFv (Juno Therapeutics). KO was measured by staining surface expression of PD1 (BioLegend).

**Genomic DNA extraction and knock-in validation**. Genomic DNA was extracted (PureLink Genomic DNA Mini Kit, Invitrogen) from modified cells expression the transgenic and integrated receptor. PCRs were performed from the 5' homology arm to 3' homology arm to verify full construct integration. PCR amplifying the DNA sequence from outside the 5' genomic DNA into the integrated DNA sequence validated the insertion into the desired loci. Purified PCR products were subsequently Sanger sequenced (Eurofins).

**In vitro characterization**. For in vitro expansion edited T cells were incubated with EBV-transformed B cells (LCL) in a 1:7 (E:T) ratio and the described protocol was performed accordingly[33,34]. CAR frequency and absolute numbers were monitored and measured over indicated time points of expansion.

Sequential stimulation experiment was performed using expanded and rested engineered CAR T cells. Expamers stimulation was done as previously described. Stimulation signal was disrupted by adding 1 mM Biotin. Cells were subsequently washed and transferred into fresh media containing low dose IL-2 (25 IU ml$^{-1}$). Surface expression of activation markers was monitored by flow cytometry (see below) and cell count was measured (Nucleocounter, Chemotec).

For in vitro killing assay, xCELLigence RTCA System (ACEA Biosciences Inc.) was used. Specific lysis was measured targeting CD19 expressing Human Embryonic Kidney cells (HEK293-CD19$^+$). $2 \times 10^4$ target cells were seeded into 96 well E-Plate (ACEA Biosciences Inc.) and rested overnight. Engineered CAR T cells were added in a 5:1 (E:T) ratio and incubated for indicated time. Changes in impedance signal were measured in real-time and analyzed using RTCA Software Pro (ACEA Biosciences Inc.). All used cell lines were unmodified and purchased from ATCC.

**Flow cytometry**. Flow cytometry was performed as previously described[33]. Following antibodies were purchased from BioLegend and used for the analysis. CD3 (clone OKT3-PC7), CD4 (clone OKT4-BV421), CD8 (clone RPA-T8-BV510), CD45 (clone HI30), PD1 (clone EH12.2H7-FITC and -BV421), CD69 (clone FN50-APC/Fire), CD19 (clone HIB19-BV421 and -FITC), and CD25 (clone BC96-BV605). In addition, an anti-idiotype (CD19) antibody produced in-house coupled to APC (Bristol Myers Squibb) and Propidium Iodide (Thermo Fisher Scientific) was used. All antibody dilutions were done according to manufacturer's recommendations.

**In vivo tumor xenograft model**. NSG-SGM3 (NOD.Cg-Prkdcscid Il2rgtmWjlTg (CMV-IL3, CSF2, KITLG) 1Eav/MloySzJ, NSGS, The Jackson Laboratory) mice were purchased from Jackson Laboratory (The Jackson Laboratory) and bred in-house in a pathogen free facility at the Technical University Munich according to standard procedures. 6–10-week-old mice (male and female) were used for experiments. All experiments were conducted as previously described and in the accordance with the guidelines of the Regierung von Oberbayern. All protocols were approved by Institutional Animal Care and Use Committee of the Regierung von Oberbayern (ROB-55.2-2532.Vet_02-17-138 and Vet_02-18-162).

B-cell rechallenge was executed by injecting $10 \times 10^6$ donor-matched B cells 2 weeks post tumor clearance. Animal experiments were executed in compliance with all relevant ethical regulations for animal testing.

**Data analysis**. Flow cytometric data were analyzed using FlowJo software (FlowJo, LLC). Graphs and statistical analysis were generated using GraphPad Prism software (GraphPad Software).

**Statistics and reproducibility**. For all the experiments sample size was selected base of standard experimental practices (mostly 3 or more independent experiments), unless number of repetitions was technically challenging e.g., mouse studies. The statistical significance was analyzed using (un)paired Student's $t$-test. No statistical samples size calculations were performed.

**Reporting summary**. Further information on research design is available in the Nature Portfolio Reporting Summary linked to this article.

## Data availability

The authors declare that data generated or analyzed for this study are available within the paper and its supplementary information. The source data behind the graphs are available in Supplementary Data 1. Described findings are proprietary of Bristol-Myers Squibb Company and some information are considered a trade secret. The raw data that support the findings of this study are available from the authors but restrictions may apply to the availability of these data. Data are, however, available from the authors upon reasonable request and with permission from the Bristol Myers Squibb Company.

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

## Acknowledgements

We thank Michaela Wagner, Eileen Benke and Dr. Katrin Manske for their outstanding technical support.

## Author contributions

S.P.F., C.T., M.E., V.C., and M.G. contributed to study design, performed experiments, and analyzed the data. K.S. and D.H.B. developed and provided gene engineering protocols. C.S., L.G., and M.P.P. designed and directed the project. S.P.F., M.E., and M.P.P. compiled the manuscript. All authors reviewed this manuscript.

## Competing interests

S.P.F., C.T., M.E., V.C., M.G., D.H.B., C.S., L.G. and M.P.P. are currently employed by Juno Therapeutics GmbH, A Bristol-Myers Squibb Company and own stocks of Bristol-Myers Squibb. L.G., C.S., and M.P.P. are listed as inventors on previously filed related patent applications. All other authors declare no competing interests.
