## [Peer Review File · Communications Biology]

Reviewers' comments:

Reviewer #2 (Remarks to the Author):

Currently, although most Chimeric Antigen Receptor T cell commercial products were widely produced with viral vector for gene delivery purpose, it still possesses many limitations. In this study, Effenberger et al. manufactured activation- inducible CAR-T cells with CRISPR-mediated targeted integration. They established a new method to manipulate the CAR-T cell function both in vitro and in vivo. By taking advantage of CRISPR/Cas gene-editing, they integrated CAR gene precisely into the target loci in the downstream of promoters. They showed that the CAR expression was successfully regulated by controlling the T cell activation. In general, this is an interesting and impactful study. It might provide an alternative novel approach to for CAR-T product manufacture.

Major concern:

- 1.The authors created PD1 KO CAR KI cells, but did not show the functional superiority of them compared with LV control cells neither in vivo nor in vitro experiments. Since the authors mentioned that knockout of PD1 can effectively prolong the survival of T cells, they should add tumor load experiments to explore whether PD1 KO CAR KI cells group mice live longer than LV control cells? Or they need edit the manuscript and remove this claim.
- 2.The authors found that CAR expression in PD1-CAR KI cells group was significantly increased after restimulation, while in TRAC-CAR KI cells group and LV control group was not significantly changed after restimulations. However, the results of CAR-T cell amplification showed that PD1-CAR KI cells and LV control cells expanded effectively after both primary and secondary stimulation. How to explain these seemly "contradictory" results?
- 3.When CD19 CAR-T cells were co-cultured with CD19 antigen expressing LCL cells, it was assumed that T cell activation would increase CAR expression. Why CAR expression declined with the increase of CAR-T cell activation after first stimulation?
4. For in vivo experiments, the proportion of CAR expression in PD1 KO CAR KI cells group dropped to almost zero after 49 days of CAR-T cells transfer. Is it possible to conduct a rechallenge experiment at this time to prove the activation and amplification ability of CAR-T cells in PD1 KO CAR KI cells group, as well as the expression of CAR?

Minor point

- 1.The rechallenge experiment in this manuscript demonstrated that the LV control cells group created more CAR-T cells and had a stronger response. Does it also prove that the LV control cells group had a more potent capacity of CAR-T cells to inhibit tumor recurrence? If this is applied for clinically purpose, is it a disadvantage of PD1 KO CAR KI cells? Please discuss this.
- 2.Besides PD1, there are other T cell exhaustion markers include Tim3 and TIGIT. Why PD1 was selected for knockout editing? What is the advantage and disadvantage?

Reviewer #3 (Remarks to the Author):

In the submitted short report, the authors present a strategy for activation dependent CAR expression (aka engineered feedback loop CAR). The authors present proof-of-concept data supporting this approach. They use CD19-CAR and integrate it in the PD-1 locus using CRISPR-Cas9 based approach followed by a quick in vitro and in vivo characterization.

Concerns:

- I believe there is a small glitch in this system. It is not clear where/when the initial T cell activation is coming from. PD-1 is not expressed in non-activated T cells. If CAR is knocked-in in that locus, it means that initially it is not expressed. KI CAR T cells are activated in antigen dependent manner but since CAR is not expressed, T cells can't be activated. How do you "start" this CAR? It will get even more complicated in vivo as it seems like one would have to activate T cell exogenously before infusion. A better approach would be to KI a gene (cytokine, chimeric switch receptor etc) that is beneficial for T cells fitness and that can be expressed in the activation-dependent manner to avoid toxicity (proof-of-concept presented in PMID: 32604839). Overall, it seems that the proposed feedback loop is hard to initiate and is short lived.

- KI efficiency in PD-1 locus is 4.17% (Fig 1C) and in CD69 locus is 11% (Supp Fig). It is not clear why the authors decided to go with PD-1 when CD69 is higher.

- It is not clear why the authors are not showing IVIS imaging post-rechallenge (Fig 1I)

- Fig 1H: one animal in each CAR treatment group disappears (die) at day28. Why?

- to demonstrate the flexibility of this approach, the authors should do another in vivo experiment with T cells where CAR is knocked-in in CD69 locus.

- a caveat of the study is that the authors tested this approach with CD19-CAR targeting CD19 which is a highly expressed antigen. Does this system still work for antigens that are expressed at much lower levels like HER2, GPC2.

Reviewers' comments:

Reviewer #2 (Remarks to the Author):

Currently, although most Chimeric Antigen Receptor T cell commercial products were widely produced with viral vector for gene delivery purpose, it still possesses many limitations. In this study, Effenberger et al. manufactured activation- inducible CAR-T cells with CRISPR-mediated targeted integration. They established a new method to manipulate the CAR-T cell function both in vitro and in vivo. By taking advantage of CRISPR/Cas gene-editing, they integrated CAR gene precisely into the target loci in the downstream of promoters. They showed that the CAR expression was successfully regulated by controlling the T cell activation. In general, this is an interesting and impactful study. It might provide an alternative novel approach to for CAR-T product manufacture.

We thank reviewer for appreciating the significance of our work.

Major concern:

1. The authors created PD1 KO CAR KI cells but did not show the functional superiority of them compared with LV control cells neither in vivo nor in vitro experiments. Since the authors mentioned that knockout of PD1 can effectively prolong the survival of T cells, they should add tumor load experiments to explore whether PD1 KO CAR KI cells group mice live longer than LV control cells? Or they need edit the manuscript and remove this claim.

We thank the reviewer for the comment and apologize for the lack of clarity. Indeed, we did not show benefit of PD1 knock-out as it would require prolonged in vivo investigations reviewer suggested. Technical challenges of these studies on our end are explained in comments below.

Moreover, we did not show the functional superiority of PD1 KO CAR KI cells over LV control. It seems we were imprecise in our argumentation. We agree that PD1 KO CAR KI cells are not functionally superior, but the novel gene editing design in itself (knock-in into locus under control of activation induced promoter) gives a new opportunity for superior control of CAR T cells in vivo. For example, T and B cell aplasia can be prevented when inactivated cells cease to express CAR and permit reconstitution of T and B cells. This is how we would define the superiority of our concept over random LV-mediated CAR integration. The manuscript has been adjusted accordingly.

2. The authors found that CAR expression in PD1-CAR KI cells group was significantly increased after restimulation, while in TRAC-CAR KI cells group and LV control group was not significantly changed after restimulations. However, the results of CAR-T cell amplification showed that PD1-CAR KI cells and LV control cells expanded effectively after both primary and secondary stimulation. How to explain these seemingly “contradictory” results?

We thank for the question. The seemingly contradictory results in Fig 1D & E can be explained by the difference in stimulus used. In Fig 1D, we took advantage of polyclonal anti-CD3/CD28 stimulation that will activate equally all the cells (i.e., cell populations with and without CAR), but should not have an impact on CAR expression when using conventional editing (TRAC KI or LV). Only in case of activation induced CAR expression we indeed observed a significant increase in CAR confirming our main hypothesis. Conversely, in Fig 1E we used antigen/CAR-specific stimulation to successfully expand only cells expressing CAR construct. In this case, all cells expressing CAR (irrespective of the type of editing used) will expand but not the ones without the CAR. Nevertheless, even here we observed higher expansion rate of PD1 KO CAR KI cells over LV control indicating synergistic effect of

CAR-specific stimulation and activation induced CAR expression. The manuscript has been adjusted for clarity.

3. When CD19 CAR-T cells were co-cultured with CD19 antigen expressing LCL cells, it was assumed that T cell activation would increase CAR expression. Why CAR expression declined with the increase of CAR-T cell activation after first stimulation?

If we understood the reviewer correctly, he is referring to Fig 1F. The apparent initial decline of CAR expression is a technical artefact of the co-culture system. When anti-CD19 CAR+ T cells interact with CD19 antigen on the surface of LCL target cells the CAR detection can be temporarily hindered due to CAR epitope masking. At later time points, the masking effect is reduced revealing a significant increase in CAR+ cells. Because of the lack of statistical significance, we didn't comment on this point previously but in agreement with the reviewer comment we now added an appropriate statement.

4. For in vivo experiments, the proportion of CAR expression in PD1 KO CAR KI cells group dropped to almost zero after 49 days of CAR-T cells transfer. Is it possible to conduct a rechallenge experiment at this time to prove the activation and amplification ability of CAR-T cells in PD1 KO CAR KI cells group, as well as the expression of CAR?

We thank the reviewer for this comment. We agree that it would be interesting to induce a re-challenge at a later time point as this would strengthen our hypothesis. Unfortunately, due to the ethical & animal welfare as well as technical limitations it is not possible for us to reliably execute the proposed experimental setting. The limitation of the used NSG mouse model is a massively increasing risk of GvHD at later time points and we are losing animals in all groups because of anticipated xenoreactivity of the transferred cells (as in Figure 1H and 1I late time points). Consequently, it is unlikely that we would have sufficient animal number to draw meaningful conclusions with incomplete groups. Therefore, in our original experiment we postponed re-challenge to the latest reasonable time point.

Minor point

1. The rechallenge experiment in this manuscript demonstrated that the LV control cells group created more CAR-T cells and had a stronger response. Does it also prove that the LV control cells group had a more potent capacity of CAR-T cells to inhibit tumor recurrence? If this is applied for clinically purpose, is it a disadvantage of PD1 KO CAR KI cells? Please discuss this.

We understand the reviewer's statement and we agree it is relevant for discussion. Indeed, in case of a potential relapse, as long as LV-transduced CAR cells are present they should control the tumor and prevent recurrence. However, constant presence of CD19 CAR T cells in patient's circulation has the downside of prohibiting any reconstitution of not only malignant but also healthy B cells leading to a chronic B cell aplasia. This is an even bigger challenge in T cell malignancies, where prolonged T cell aplasia will end up in severe uncontrolled infections. Therefore, we don't intend to discredit the use of drug products constantly expressing CAR construct, but rather highlight the new opportunities of our approach that can be beneficial under specific circumstances. For example, iPSC derived T/NK cells are very suitable for re-dosing (addressing potential relapse) but may benefit from an additional level of safety offered by our system. Appropriate statements were added to the discussion.

2. Besides PD1, there are other T cell exhaustion markers include Tim3 and TIGIT. Why PD1 was selected for knockout editing? What is the advantage and disadvantage?

We thank the reviewer for this important comment. We added TIM3 and TIGIT expression kinetic in updated Figure 1B to show their potential when applied in our concept. To our knowledge these three markers truly define exhaustion and all three are expressed at the same time. That is why we believe that targeting one is sufficient and we prioritized PD1 as it is best characterized and its inhibition has been already extensively tested in the clinic (e.g., checkpoint inhibitor cancer therapy) (Doetsch et al., 2023). Moreover, as now presented in the updated Figure 1B, TIM3 and TIGIT were not upregulated sufficiently upon activation to be ideal targets in our proof-of-concept study. The manuscript and figures were modified accordingly.

Reviewer #3 (Remarks to the Author):

In the submitted short report, the authors present a strategy for activation dependent CAR expression (aka engineered feedback loop CAR). The authors present proof-of-concept data supporting this approach. They use CD19-CAR and integrate it in the PD-1 locus using CRISPR-Cas9 based approach followed by a quick in vitro and in vivo characterization.

Concerns:

- I believe there is a small glitch in this system. It is not clear where/when the initial T cell activation is coming from. PD-1 is not expressed in non-activated T cells. If CAR is knocked-in in that locus, it means that initially it is not expressed. KI CAR T cells are activated in antigen dependent manner but since CAR is not expressed, T cells can't be activated. How do you "start" this CAR? It will get even more complicated in vivo as it seems like one would have to activate T cell exogenously before infusion. A better approach would be to KI a gene (cytokine, chimeric switch receptor etc) that is beneficial for T cells fitness and that can be expressed in the activation-dependent manner to avoid toxicity (proof-of-concept presented in PMID: 32604839). Overall, it seems that the proposed feedback loop is hard to initiate and is short lived.

We thank the reviewer for the comment as it points out that we didn't introduce our concept sufficiently. Yes, it is correct to assume that CAR will not be expressed without stimulation. However, stimulation does not have to be mediated exclusively via CAR. In many if not all manufacturing processes (including the one we envision), one of the first steps is to activate T cells using polyclonal anti-CD3/anti-CD28 stimulation. This step is needed to induce cell expansion as well as improve gene editing efficiency. We also showed that this step is sufficient to induce CAR expression in an activation dependent manner. Freshly activated CAR-expressing cells can be readily transferred in vivo (as has been demonstrated in multiple ultra-short manufacturing processes (e.g., Novartis T-Charge or Gracell FasTCAR platform). After transfer, the positive feedback loop between CAR and its antigen will perpetuate CAR expression as long as tumor is present. The manuscript has been modified for clarity.

We also thank the reviewer for the very useful suggestion of the literature reference. Indeed, Ode et al. presented a related concept of an inducible system when knocking-in IL15 into IL13 locus. Their results support the validity of our idea but also underline the importance of an appropriate locus selection. We included this reference in the manuscript and highlighted its findings.

- KI efficiency in PD-1 locus is 4.17% (Fig 1C) and in CD69 locus is 11% (Supp Fig). It is not clear why the authors decided to go with PD-1 when CD69 is higher.

We acknowledge the reviewer's comment. It is correct that KI into PD1 locus reaches 4.17% after initial stimulation. This needs to be compared with KI into CD69 locus after initial stimulation which is at 4.85% (Suppl. Fig. 1B; middle panel). In this case, a 0.68% difference can be explained by experimental variation. Moreover, not only initial KI efficiency was considered when selecting PD1 over CD69. PD1 has a well described negative effect on T cell function, contrary to CD69 which

upregulation is required for T cell differentiation and tissue retention (Cibrián et al., 2017). Therefore, PD1 KO may have an additional benefit enhancing function of CAR T cells (McGowan et al., 2020), whereas CD69 may have more of a negative effect. The target selection justification has been improved in the manuscript.

- It is not clear why the authors are not showing IVIS imaging post-rechallenge (Fig 1I)

We thank the reviewer for the comment. The images post-rechallenge are now included. For optimal display, an alternative representative experiment in Figure 1H was displayed to match with Figure 1I.

- Fig 1H: one animal in each CAR treatment group disappears (die) at day28. Why?

In original Figure 1H on day 28 all animals were still alive, but the visual grouping between IVIS images might not been perfect. Some of the mice died only at later time points due to a GvHD when kept for extended period (a common phenomenon of the used NSG mouse model). An alternative representative experiment is now displayed for better clarity and better alignment with Figure 1I.

- to demonstrate the flexibility of this approach, the authors should do another in vivo experiment with T cells where CAR is knocked-in in CD69 locus.

We agree with the reviewer that showing in vivo data for CD69 will demonstrate flexibility of our approach. Additional panels displaying appropriate in vivo experiment were added to Supplementary Figure 1. Interestingly, CAR KI into CD69 locus exerted tumor control but resulted in reduced anti-tumor efficacy compared to KI into PD1 locus and the animals had to be sacrificed already at day 28.

- a caveat of the study is that the authors tested this approach with CD19-CAR targeting CD19 which is a highly expressed antigen. Does this system still work for antigens that are expressed at much lower levels like HER2, GPC2.

We agree with the interesting point raised by the reviewer. Indeed, it would be curious to see if our setting is working as efficiently with lower density antigens. Unfortunately, as an R&D department within Bristol Myers Squibb company we have limited access to CAR construct not included in the commercial portfolio and cannot address this well taken point at the moment.

Theoretically, it might be possible to counterbalance low antigen density with increased CAR surface expression under control of a strong endogenous promotor. However, too high CAR expression can result in antigen-independent signaling leading to T cell exhaustion (Frigault et al., 2015). Another approach might be affinity maturation to enable efficient CAR binding to low-density targets. Unfortunately, high affinity CARs might tend to increased trogocytosis which will impact serial killing function (Watanabe et al., 2018 & Caserta et al., 2010). These considerations we hope to address in future manuscript. The manuscript has been modified accordingly.

References:

- Cibrián D, Sánchez-Madrid F. CD69: from activation marker to metabolic gatekeeper. *Eur J Immunol.* 2017 Jun;47(6):946-953. doi: 10.1002/eji.201646837. PMID: 28475283; PMCID: PMC6485631.
- McGowan E, Lin Q, Ma G, Yin H, Chen S, Lin Y. PD-1 disrupted CAR-T cells in the treatment of solid tumors: Promises and challenges. *Biomed Pharmacother.* 2020 Jan;121:109625. doi: 10.1016/j.biopha.2019.109625. Epub 2019 Nov 13. PMID: 31733578.
- Frigault MJ, Lee J, Basil MC, Carpenito C, Motohashi S, Scholler J, et al. Identification of chimeric antigen receptors that mediate constitutive or inducible proliferation of T cells.

Cancer immunology research. 2015; 3(4):356–67. <https://doi.org/10.1158/2326-6066.CIR-14-0186> PMID: 25600436

- Watanabe K, Kuramitsu S, Posey Jr A, June C Expanding the Therapeutic Window for CAR T Cell Therapy in Solid Tumors: The Knowns and Unknowns of CAR T Cell Biology. *Front. Immunol.*, Volume 9 - 2018 | <https://doi.org/10.3389/fimmu.2018.02486>
- Caserta S, Kleczkowska J, Mondino A, Zamoyska R. Reduced functional avidity promotes central and effector memory CD4 T cell responses to tumor-associated antigens. *J Immunol.* (2010) 185:6545–54. doi: 10.4049/jimmunol.1001867
- Dötsch S, Svec M, Schober K, Hammel M, Wanisch A, Gökmen F, Jarosch S, Warmuth L, Barton J, Cicin-Sain L, D'Ippolito E, Busch DH. Long-term persistence and functionality of adoptively transferred antigen-specific T cells with genetically ablated PD-1 expression. *Proc Natl Acad Sci U S A*. 2023 Mar 7;120(10):e2200626120. doi: 10.1073/pnas.2200626120. Epub 2023 Feb 28. PMID: 36853939.

REVIEWERS' COMMENTS:

Reviewer #2 (Remarks to the Author):

Thanks for the detailed response to the questions, the activation-induced CAR-T cells described in the article are beneficial to the normal development of T and B cells after immunotherapy. This concept has a certain innovation, but there is a lack of experimental verification at present. Is there any strong data to support this conclusion?

Reviewer #3 (Remarks to the Author):

All my comments were addressed, and I have no additional concerns.

REVIEWERS' COMMENTS:

Reviewer #2 (Remarks to the Author):

Thanks for the detailed response to the questions, the activation-induced CAR-T cells described in the article are beneficial to the normal development of T and B cells after immunotherapy. This concept has a certain innovation, but there is a lack of experimental verification at present. Is there any strong data to support this conclusion?

We thank reviewer for the positive feedback. Indeed, in the current manuscript we do not provide a strong data supporting this concept. Therefore the strength of the statements referring to T and B cell development after immunotherapy has been adjusted.

Reviewer #3 (Remarks to the Author):

All my comments were addressed, and I have no additional concerns.

We thank reviewer for the feedback and we are happy that we were able to address all the questions.